# Experimental Study on Freezing and Thawing Cycles of Shrinkage-Compensating Concrete with Double Expansive Agents

**DOI:** 10.3390/ma13081850

**Published:** 2020-04-15

**Authors:** Jinjun Guo, Ting Guo, Shiwei Zhang, Yan Lu

**Affiliations:** School of Water Conservancy Engineering, Zhengzhou University, Zhengzhou 450001, China; guojinjun@zzu.edu.cn (J.G.); guotinghs@163.com (T.G.); zsw2060502@163.com (S.Z.)

**Keywords:** shrinkage compensating concrete, double expansive admixture, rapid freezing and thawing cycles test, frost resistance, relative dynamic modulus of elasticity, mass loss rate

## Abstract

The freezing and thawing of construction concrete is becoming an increasingly important structural challenge. In this study, a shrinkage-compensating concrete based on a double expansive admixture was developed and its frost resistance was assessed through rapid freezing and thawing cycling. The frost resistance of the concrete was derived through the measurement and calculation of the relative dynamic modulus of elasticity (RDME) and the mass loss rate (MLR), and the freezing- and thawing-cycle microstructures and products of concretes with different expansive agents were analyzed using scanning electron microscopy (SEM). It was shown that changes in the properties of the concrete under freezing and thawing could be divided into three stages: slow-damage stage, fast-damage stage, and stable stage. Compared to concrete without an expansive agent, a single-expansive-agent concrete demonstrated excellent frost resistance during the slow-damage stage, but the frost resistance rapidly decreased during the fast-damage age. After 150 cycles (the stable-damage stage), the concrete with a U-type expansive agent (UEA): MgO expansive agent (MEA) mix proportion of 2:1 had the best frost resistance, with RDME and MLR values 17.35% higher and 25.1% lower respectively, than that of an expansive-agent-free concrete. These test results provide a basis for the study of frost resistance in large-scale hydraulic concrete structures.

## 1. Introduction

The high degree of workability, excellent construction performance, and economic benefits of concrete have long made it a widely used construction material solution for civil and water engineering [1]. Over the past few years, however, the freezing and thawing of hydraulic concrete in cold areas has become an increasingly challenging issue, especially in cold areas. Hydraulic concretes with low water-to-cement ratios, large dosages of cementitious material, high sand ratios, and relatively small coarse aggregate particle sizes are prone to cracking from significant autogenous and plastic shrinkage, which contribute significantly to declining frost resistance. Because water and other hazardous substances can easily ingress into concrete through such concrete cracks, the durability and frost resistance of the concrete deteriorates, threatening the security of the hydraulic structure and reducing its service life [2]. In the United States, the annual economic loss caused by frost damage is more than 6 billion dollars, while in northeast China nearly 100% of concrete buildings have been damaged by freezing and thawing [3]. An example of this is the Yunfeng Hydropower Station, which, despite being in operation for less than 10 years, has incurred freezing- and thawing-induced damage to 10,000 m^2^ of the overflow dam surface area, with an average denudation depth exceeding 10 cm [3]. Therefore, developing a durable hydraulic concrete characterized by good frost resistance properties is key to implementing temperature-cycle-resistant hydraulic structure management practices.

The frost resistance of concrete is influenced by many factors, including the water-cement ratio, compactness of the concrete, pore structure, etc. Many cold regions, including northern Europe, North America, and the former Soviet Union, have attached great importance to frost resistance starting as early as the 1940s. Conventional methods for improving the frost resistance of concrete include the reduction of the water-cement ratio [4], special curing, [5] and the addition of additives [6]. The additive method, which is preferred by many researchers, includes the use of porous lightweight aggregate [7], supplementary cementitious materials (e.g., alkali, fly ash, lime powder) [8,9], fibers [10], expansive agents [11], etc. However, the use of lightweight instead of natural aggregates decreases the strength of the concrete, while the high costs of cementitious materials and fibers can mitigate against their use in large water projects. In this context, the addition of expansive agents is an effective solution for the effective compensation of concrete shrinkage to reduce the generation of cracks. The shrinkage compensation occurs as a result of the difference in density between the expansive agent and its hydration products, which can double in volume when the expansive agent reacts with water to convert into other products. Through this mechanism, the pore structure of the concrete is improved as a result of the refinement of pores, improvement of pore size and morphology, and decrease in porosity. Given this, a large number of researchers have studied the performance of expansive agents [12,13]. For example, expansive concretes based on the formation of ettringite from calcium sulfoaluminate agents were presented as an effective alternative to minimize the problems derived from shrinkage and low tensile strength at the early age of the concrete in [14]. Li et al. performed an experimental investigation on the shrinkage and mechanical properties of concrete compared to the self-compacting concrete (SCC) with or without an expansive agent, and the results show the calcium-sulfoaluminate expansive agent with content optimized to be 10% mass of binders [15]. Roller compacted concrete (RCC) with different MgO contents was developed to evaluate the improvement of crack resistance in [16], which show that the MgO can improve the crack resistance of RCC and its mechanical properties in the long term. The effects of MEA and fly ash dosage in cement replacement were investigated by Sherir et al. with the results indicating that 5% MEA should be used in concrete self-healing systems to enable the healing of microcracks without affecting durability [17]. Wyrzykowski et al. showed that a combination of expansion-promoting (based on calcium sulfoaluminate (CSA)) and shrinkage-reducing additives (based on superabsorbent polymers (SAP) and shrinkage-reducing admixture (SRA)) could be used to produce concrete with outstanding mechanical properties and very high frost resistance at early ages [18]. Mo et al. revealed that the addition of expansive agents can result in the formation of the chemically stable products considered to be the primary components for the effective compensation of concrete autogenous shrinkage at specific ages [19]. Sung et al. used a CSA-based expansive additive to compensate for the shrinkage of alkali-activated material (AAM) mortar, resulting in an expansive agent with an excellent compensation effect on drying shrinkage at early ages, although the long-term effect was minimal [20]. In Na et al.’s study, it was found that early-age frost resistance can be improved through the use of CSA-based expansive agents, which reduce or even prevent the formation of microcracks, although the effect was not obvious in later periods [21]. However, most such studies have focused on the improvement of frost resistance at a specific application age instead of focusing on all ages.

The results of these previous studies reveal that adding only a single expansive agent to concrete results in limited shrinkage compensation. For example, CSA-based expansive agents such as UEA can compensate for only the early shrinkage of concrete but have little effect on later-stage compensation. Other expansive agents, such as MEA, are characterized by delayed expansion, which has a good effect on compensating the later-stage shrinkage of mass concrete. However, MEA concretes show higher initial shrinkage and are subject to continuous shrinkage over the long term, which can result in numerous cracks. Most hydraulic concrete problems arise from shrinkage cracking, which not only destroys the integrity of concrete but also worsens a number of concrete properties such as frost resistance and impermeability. Given growing requirements in terms of hydraulic concrete structure quality and service life, single expansive agents that are only effective at a specific age have limited functionality. This makes it necessary to develop compound expansive agents that exhibit large shrinkage in the early ages but slight shrinkage over the long term to improve the frost resistance of the concrete and extend its service life. 

The objective of this study is to develop a double expansive resource admixture based on UEA and MEA and analyze its effects on the frost resistance of concrete structures. Four different UEA: MEA mix proportions (4:1, 2:1, 1:1, and 1:0) were subjected to rapid freezing and thawing cycling tests. The effects of various expansive agent percentages on frost resistance were explored by measuring the relative dynamic moduli of elasticity (RDME) and mass loss rate (MLR) of the respective concretes. The microstructures of the shrinkage-compensating concretes were imaged using a scanning electronic microscope (SEM). Based on these results, an optimal compound ratio of the two expansive agents was proposed to provide a theoretical basis for the development of relevant test procedures for concrete subjected to freezing and thawing cycles.

## 2. Materials and Methods

### 2.1. Materials

The cement used in this study was ordinary Portland cement 42.5R, which conforms to Chinese standard GB175-2007 [22]. The coarse aggregate was limestone macadam with a density of 2760 kg/m^3^, which was composed of two particle types with different diameters: 40% of the aggregate had a particle diameter of 5–20 mm; 60% had a particle diameter of 20–40 mm. Natural river sand with a fineness modulus (FM) of 2.8 and density of 2640 kg/m^3^ was used as the fine aggregate. Type I fly ash (FA) at a dosage of 20% was used as mineral additive; the properties of fly ash are listed in Table 1. As a superplasticizer, FDN-C at a dosage of 2.2%, for which the properties are listed in Table 2, was used. Common tap water was used as mixing water.

UEA and MEA were used to develop the shrinkage compensating concrete. The mechanism of these expansive agents can be explained in terms of their respective hydration reactions, which are given by the following equations [10,21,23]: (1)$$\mathrm{UEA}:\mathrm{CA}+3{\mathrm{CaSO}}_{4}+2\mathrm{Ca}{\left(\mathrm{OH}\right)}_{2}+30H_{2}O\to C_{3}A\cdot 3{\mathrm{CaSO}}_{4}\cdot 32H_{2}O$$
(2)$$\mathrm{MEA}:\mathrm{MgO}+H_{2}O\to \mathrm{Mg}{\left(\mathrm{OH}\right)}_{2}$$
in which ettringite (C_3_A·3CaSO_4_·32H_2_O) and Mg(OH)_2_ are produced by UEA and MEA hydration, respectively. In the study, the formation hydrates were used to fill the pores to improve the pore structure and make the concrete highly compact. The properties of the UEA and MEA are listed in Table 3 and Table 4, respectively.

### 2.2. Concrete Mixes

A shrinkage-compensating concrete with a strength grade of C25 was used to investigate the effects on frost resistance of incorporating various volume fractions of UEA and MEA. A detailed mix design of the UEA and MEA and their mineral additives is given in Table 5. According to GB/T 1346-2011 [24], if the content of expansive agent is about 10%, it can improve the performance of concrete best. Therefore, the total mass fraction of UEA and MEA added into concrete is 10% in this experiment. Moreover, it is indicated that the highest content of UEA is 10%, while that of MEA is 5%. So in the experimental program, the concrete with 10% UEA was developed, and the highest mass fraction of MEA (5%) was also used to develop concrete. Except these two concretes, two groups of concrete with different mix propositions of MEA and UEA were also developed. The mass fraction of UEA in a group of concrete is twice that of MEA, while the mass of UEA in the other group of concrete is four times that of MEA. Therefore, five mix designs—a plain concrete (PC control) and four shrinkage compensating concretes, called UM_10_, UM_41_, UM_21_, and UM_11_—with mixture ratios of 1:0, 4:1, 2:1, and 1:1, respectively, were developed. 

### 2.3. Experimental Procedure

The freezing and thawing cycles test was generally used to simulate the damage to buildings caused by freezing and thawing, because of its many advantages, such as simulating the real damage environment, shortening the experiment time. Furthermore, the results obtained through the freezing and thawing cycles test can be used to effectively guide the engineering practice. Therefore, a rapid freezing and thawing cycling test was performed in accordance with GB/T 50082-2009 [25]. The tested specimens were prism shaped and had the dimensions 100 mm × 100 mm × 400 mm. After 24 d of standard curing (the temperature is (20 ± 2) °C and the humidity is higher than 95%), the specimens were cured in water for an additional 4 d and then, after the initial mass and fundamental transverse frequency of the concrete were measured, the freezing and thawing cycles were applied. Within each cycle, the temperature of the specimens was lowered from 5 to −18 °C and raised again to 5 °C over approximately 4 h. The mass and fundamental transverse frequencies of each specimen were measured every 25 cycles; the procedure used to determine the fundamental transverse frequency is shown in Figure 1. RDME and MLR were used as the primary indicators for evaluating the frost resistance of the concrete. According to the GB/T 50082-2009 [25], and referring to literatures about frost resistance of concrete [26,27], the RDME can be calculated as follows:(3)$$P_{i}=\frac{f_{ni}^{2}}{f_{0i}^{2}}\times 100\%$$
where *P_i_* is the relative dynamic modulus of elasticity after n cycles of freezing and thawing, *f_n_* (in Hz) is the fundamental transverse frequency after n cycles of freezing and thawing, and *f*_0*i*_ (Hz) is the fundamental transverse frequency at the beginning of the freezing and thawing process. A higher relative dynamic modulus of elasticity corresponds to better frost resistance.

The MLR of the shrinkage-compensating concrete is calculated as follows:(4)$$\Delta W_{ni}=\frac{W_{ni}-W_{0i}}{W_{0i}}\times 100\%$$
where Δ*W_ni_* is the mass loss rate of the concrete after n cycles of freezing and thawing, *W_0i_* (in g) is the mass of the concrete at the beginning of the freezing and thawing process, and *W_ni_* (g) is the mass of the concrete after n cycles of freezing and thawing. A smaller mass loss rate corresponds to better frost resistance.

### 2.4. SEM Analysis

The microstructures of the concretes were imaged with a SEM every 25 cycles. The SEM analysis was used to explore the correspondence between the microstructural changes and the concrete macroscopic properties. Using the gray scale differences in the image, C–S–H, hydration products such as ettringite and Mg(OH)_2_ could be distinguished. 

In this experiment, a core sample was drilled through the concrete core drilling machine from the concretes that have subjected to freezing and thawing cycles, which was used to conduct SEM test. The specific sampling procedures are as follows.

(1)Drill core sample: a concrete core drilling machine was used to drill a core sample with a diameter of 20 mm and a height of 100 mm at the center of the bearing surface of the concrete specimen, as shown in Figure 2a.(2)Cut thin slices: the thin slices with a thickness of 2 mm were cut at different longitudinal positions (5 mm, 10 mm, 15 mm, 20 mm, 25 mm, and 30 mm) from the erosion surface of the cylindrical core sample by using a cutter, as shown in Figure 2b.(3)Spray gold: the thin slices with a thickness of 2 mm that were cut at different longitudinal positions from the erosion surface of the cylindrical core sample in step (2) were sprayed gold. First, the cut samples were placed in an energy-saving box type electric furnace for drying about 12 h. It is considered that the drying was completed when the quality of the samples no longer changes. After the sample surface was dried, the sample was placed in an SBC-12 type ion sputtering apparatus to spray gold to the sample. The sample that was sprayed gold is shown in Figure 2c.(4)Observe: After the gold spray was completed, the sample was placed in the scanning electron microscope sample chamber for observation. Observations were conducted in an airtight reactor without adding O_2_, and other carrier gas.

## 3. Results

### 3.1. Analysis of Frost Resistance of Concrete

Each concrete sample was subjected to 150 freezing and thawing cycles, with the analysis results revealing developmental trends that differed by percentages of UEA and MEA. The RDME and MLR results over the course of the freezing and thawing cycles are shown in Figure 3 and Figure 4, respectively. It is seen that, over the course of the cycling, the RDME results for the various types of concrete declined (Figure 3) while the MLR increased (Figure 4). From Figure 3, the RDMEs of respective concretes first decrease slowly, then decrease rapidly, and also finally change a little. At the same time, the change of MLRs also undergoes three stages characterized by slow increase, fast increase and slow development. Moreover, from this figure, after 50 cycles, the RDME of UM_10_ was higher than the other samples, and the MLR was lower, which show that the performance was better. However, with the development of freezing and thawing cycles, the RDME of UM_10_ decreases fast, with a rate of 16.2%. After 75 cycles, the rate of change of RDMEs of UM_11_, UM_41_, and UM_21_ were 8.3%, 10.9%, and 7.4%, respectively. It is shown that except PC control, the change of RDMEs of all samples was high between the 50 cycles and 75 cycles. But after 75 cycles, with the development of cycles, the change of RDMEs was slowly decreasing. The change of MLRs was same as RDMEs. Therefore, based on these RDMEs and MLRs patterns, the process of freezing and thawing cycling can be divided into three stages: (1) A slow-damage stage during cycles 0–50; (2) a fast-damage stage from cycles 50–75; and (3) a stable stage after cycles 75. The RDME and MLR patterns for the respective concretes during each stage were then analyzed separately.

#### 3.1.1. Slow-Damage Stage

During this stage, the RDMEs of the shrinkage-compensating concretes decreased slightly while the MLRs changed slightly. It is seen from Figure 3 that the RDMEs of the concretes with expansive agents were more than 85% during the slow-damage stage and decreased by 9.51% on average, while the RDME of the PC control was less than 80%. As shown in Figure 4, the PC control had a MLR of 2.08%, while the MLRs of the shrinkage-compensating concretes were all less than 1%, with an average mass loss rate of 0.97%, which indicates better frost resistance than PC control. The improvement in the frost resistance as a result of the presence of expansive agent can be explained as follows. In previous studies, it was shown that the rapid development of the UEA hydration reaction in the shrinkage-compensating concrete is essentially completed within 7 d of aging; by contrast, the MEA hydration proceeds very slowly during the early ages, with only 57% of the MEA forming Mg(OH)_2_ crystals within the first 180 d of hydration [28]. Therefore, during the early freezing and thawing cycles, the UEA used in this experiment played a leading role in enhancing the concrete’s frost resistance. The hydration of the UEA compensated the drying shrinkage of concrete during the early ages to reduce the generation of cracks. On the other hand, the ettringite crystals (C_3_A·3CaSO_4_·32H_2_O) generated by UEA hydration filled the internal pores to improve the compactness of the concrete. Therefore, when the water in the pores underwent freezing and thawing cycles, small volume changes occurred in the concrete owing to the expansion of the expansive agents, resulting in an improved frost resistance in the shrinkage-compensating concrete relative to that of the PC control. 

As shown in Figure 3 and Figure 4, after 50 cycles the RDME values of the respective concretes were, in the order from highest to lowest, UM_10_ > UM_41_ > UM_21_ > UM_11_ > PC, while the MLRs were, in the order from highest to lowest, PC > UM_11_ > UM_21_ > UM_41_ > UM_10_. These results can be attributed to a number of factors. It has been shown that frost resistance is significantly improved at a UEA mass fraction of about 10% and that, within this range, the frost resistance is positively correlated with the UEA mass fraction [29]. The ettringite crystals generated by UEA hydration fill the pores and are strengthening the concrete in this stage and deteriorate concrete after the tension pressure is exceeded. In this study, the mass fraction of UEA of UM_10_ was 10%, in which most of pores were filled by ettringite crystals produced by hydration reaction of UEA. Therefore, the frost resistance of UM10 with mass fraction of UEA of 10% was the best, with the highest RDME of 91.51% and the lowest MLR of 0.24%. Moreover, the mass fractions of the UEA in the UM_41_, UM_21,_ and UM_11_ samples were 8, 6.6, and 5%, respectively, and the RDMEs of these concrete were 91.51, 90.30, and 87.07% and the MLRs were 0.83, 1.14, and 1.40%, respectively. The RDMEs of these concrete samples were all higher than the RDME of PC control, which indicates that the frost resistance is better than PC control. That is to say, since the mass fractions of UEA in these concretes, the frost resistance of concrete was improved in different degree.

#### 3.1.2. Fast-Damage Stage

As the freezing and thawing cycles continued, the RDMEs of the concretes decreased rapidly at an average rate of 13.65%. It is seen from Figure 3 that the MLR of the PC control increased to 4%, during the fast-damage stage, while the MLRs of the expansive-agent concretes were around 3.5%, with a rate of increase of 2.04%. These decreases in RDME and increases in MLR can be attributed to the fact that most of the UEA hydration had been completed and the remaining UEA had slow reactivity, while the rate of MEA hydration was also slow. The volume of Mg(OH)_2_ generated by MEA hydration was also small and had a slow growth rate. As a result, the compactness of the concrete was reduced during the fast-damage age and the pore structure was not effectively improved, resulting in rapid decreases in RDME after 75 freezing and thawing cycles. Furthermore, many microcracks had already formed in the concretes during the freezing and thawing cycles of the previous age. As the cycles continued, the microcracks connected and expanded to form larger cracks. After many cycles, the mortar and gravel at the crack edges gradually fell off, resulting in rapid increases in MLR.

It is seen from Figure 3 that the RDME of UM_10_ changed most significantly and decreased the fastest, with an average rate of decline of 20.02%. After 75 cycles, the RDMEs of the respective concretes were, in the order from highest to lowest, UM_21_ > UM_41_ > UM_11_ > UM_10_ > PC, while the MLRs were, in the order from highest to lowest, PC > UM_10_ > UM_11_ > UM_41_ > UM_21_. These results can be attributed to the fact that, by this age, the UEA hydration reaction in concrete has been essentially completed but there was a lack of other hydration products to continually improve the pore structures of the concretes. As a result, the structures were not effectively improved, and the compactness of the concrete remained low. As the amount of Mg(OH)_2_ produced by MEA hydration increased, the compactness of the MEA concrete also increased, resulting in an enhanced frost resistance relative to the PC control and UM_10_ counterpart. It has been suggested that the frost resistance of concrete is best at an MEA mass fraction of about 3% and that excessive dosage of MEA negatively affects the frost resistance [13]. This corresponds to the results achieved in this study, in which the RDME of the UM_21_ was highest among the concretes while its MLR was lowest. 

#### 3.1.3. Stable Stage

As the number of freezing and thawing cycles progressed beyond 75, the RDMEs of the concretes decreased slowly at an average rate of 3.84%, while the MLRs increased slowly at an average rate of only 1.13%. After 150 cycles, the RDME of the PC control was lower than 60% and the MLR was more than 5%. According to relative standards, concrete will lose its applicability when its RDME is lower than 60% or the MLR is above 5%, indicating that the PC control had become useless by the stable age. From Figure 3, it is seen that the stable-age RDMEs of UM_21_, UM_41_, and UM_10_ were 78, 75, and 71%, respectively, while the RDME of UM_10_ was 68%. As shown in Figure 4, the MLRs of the concretes with compound-expansive agents were all below 5%, indicating that the frost resistances of the concretes mixed with compound-expansive agents were all superior to those of the concrete with the single expansive agent.

After 150 cycles, the RDMEs of the respective concretes were, in the order from highest to lowest, UM_21_ > UM_41_ > UM_11_ > UM_10_ > PC, while the MLRs were, in the order from highest to lowest, PC > UM_10_ > UM_11_ > UM_41_ > UM_21_. These results confirm that the frost resistances of the concretes with UEA and MEA were superior to those of the concrete without an expansion agent. As the freezing and thawing cycles continued beyond 100, the UEA hydration reaction completed while the MEA hydration continued. The Mg(OH)_2_ generated by the MEA hydration process continued to fill into the internal pores of the concretes, effectively reducing the porosities and improving the pore structures, which in turn enhanced the frost resistance capabilities of the concretes. Mo et al. found that the frost resistance of concrete is optimized at an MEA mass fraction of about 3% and those excessive MEA dosages can negatively affect the frost resistance [30]. The mass fraction of MEA in the UM_21_ sample was 3.3%. Most of the pores in the UM_21_ concrete were filled by Mg(OH)_2_ particles produced by the hydration of MEA, resulting in a high degree of compactness. As a result, the RDME of UM_21_ was the highest among the concretes and its frost resistance was the best. The mass fraction of MEA in UM_41_ was 2.5%, corresponding to a frost resistance worse than that of UM_21_ but better than those of the others, although all compound-expansive samples had, to some degree, enhanced frost resistances. The mass fraction of MEA in the UM_11_ sample was 5%, far higher than the 3% optimum identified in [30]. This excess in MEA content had a negative influence on the performance of the UM_11,_ and resulted in the formation of numerous cracks in the internal structure of the concrete, a result that can potentially account for the low RDME of UM_11_ relative to UM_21_ and UM_41_. It is also important to note that, while the UEA was most effective at improving the frost resistance during the earlier ages, the frost resistance enhancement relative to the PC control persisted during the late stages as a result of the reduced porosity caused by the filling effect of the ettringite hydrates.

An analysis of the respective concretes indicates that, after 50 cycles, the RDME and MLR values of the 1:0 UEA: MEA mixture was high and low, respectively, indicating an excellent compensation effect against frost resistance during the slow-damage age. Over the longer term, however, this effect diminished as the number of freezing and thawing cycles increased, preventing further improvement in the frost resistance. By contrast, the 2:1 UEA: MEA mix was characterized by moderate frost resistance during the slow-damage age, which effectively improved after 75 cycles, with increased RDME and reduced MLR values. It was found that the addition of an appropriate UEA could compensate shrinkage and improve the frost resistance of concrete during early ages but that the effect was reduced from a longer-term perspective; by contrast, the frost resistance of the concretes with double expansive resources admixture was better from a long-term perspective. These results indicate that the addition of UEA and MEA in appropriate proportions can effectively improve the frost resistance of concrete.

### 3.2. SEM Analyses of Different Concretes

SEM analyses were conducted on preloaded cracked concrete prism specimens with various compositional makeups. For the assessment, three expansive agent mass fractions were selected—UM_10_, UM_21_, and the PC control—to represent the concretes with compound expansive agents, single expansive agents, and no expansive agents, respectively. Figure 5 shows a microscopic image of the early-age specimens (after 50 freezing and thawing cycles). A large amount of C–S–H is seen in the PC control microstructure as a result of hydration in the cement (Figure 5a). Microcracks are also observed in the PC control structure, indicating that it has been affected by freezing and thawing. By contrast, there are no obvious cracks in the UM_10_ microstructure (Figure 5b), in which, in addition to C–S–H produced by cement hydration, there is an increased presence of needle-like ettringite crystals produced by UEA hydration. Most of the pores in the UM_10_ are filled with ettringite crystals and C–S–H, which effectively reduce the porosity and increase the compactness of the concrete. From a macro perspective, this result supports the high frost resistance of UM_10_ during the early ages. The microstructure of UM_21_ (Figure 5c) also contains short needle-like ettringite crystals produced by UEA hydration. The main difference between UM_10_ and UM_21_ in terms of microstructure is the content of ettringite crystals; compared with the UM_10_, the porous interior of the UM_21_ allows for the freezing of high amounts of water, potentially leading to more pronounced frost heaving force as a result of water freezing. This indicates that improvements in early-age frost resistance can be achieved by changing the mass fraction of UEA.

Figure 6 shows microscopic images of the samples in the mid-phase of freezing and thawing cycling (after 75 cycles). As the freezing and thawing cycles progress, the cement in the PC control continues to undergo hydration reaction, producing a large amount of C–S–H sheet cementitious materials (Figure 5a). After 75 cycles of freezing and thawing, the microcracks in the PC control have gradually widened and an increasing number of cracks formed (Figure 6a). In the concrete pores and surrounding areas of the UM_10_, there are many needle bar and plate strip ettringite crystals, most of which are larger than those in the previous age (Figure 6b). Microcracks have also formed in the UM_10_ microstructure. In the UM_21_ microstructure, ettringite crystals have grown gradually, a small number of Mg(OH)_2_ particles have been deposited in the concrete pores as a result of MEA hydration, and no obvious cracks are present. These features help to explain why the frost resistance of concrete mixed with double expansive admixture is superior to those of the UM_10_ counterpart and PC control.

Figure 7 shows microscopic images of the later stages of freezing and thawing (after 125 cycles). As the freezing and thawing cycles proceed, the number and size of cracks in the PC control increase (Figure 7a). An increasing number of microcracks are also observed in the UM_10_, which gradually widen and accumulate into larger clusters (Figure 7b). As the number of freezing and thawing cycles increases, MEA hydration continues and a large amount of delayed-formation Mg(OH)_2_ is produced. A large amount of Mg(OH)_2_ and ettringite crystallization is also observed in the pores of the UM_21_ (Figure 7c), which effectively improves the pore structure and the compactness of the concrete. However, the UM_21_ has fewer cracks than the other samples, suggesting that the formation of delayed MEA hydration product features with cementitious characteristics might be a solution for compensating the weaknesses incurred by the rapid hydration of UEA during the early stages and, as a result, improving the frost resistance of the concrete.

In conclusion, the micro morphologies of concretes with different expansive agents varied significantly over the course of repeated freezing and thawing cycles. The size and number of cracks in the PC control increased, leading to the loss of the application properties of the concrete. In the early-stage UM_10_, UEA hydration led to the creation of numerous ettringite crystals, which filled internal pores and enhanced the frost resistance. After 125 cycles, ettringite crystals produced by UEA hydration were aggregated into large clusters and microcracks were formed in the concrete. In the UM_21_, small-needle ettringite crystals produced by UEA hydration filled some of the pores during the early stages, leading to sub-optimal frost resistance. As the number of cycles increased, the MEA reactivity slowed and the number of Mg(OH)_2_ particles produced by the MEA hydration increased only gradually. After 125 cycles, a large quantity of delayed-formation Mg(OH)_2_ filled the pores and cracks of the concrete, improving its compactness. As a result, the UM_21_ had the least amount of observable microcracks, indicating an improved frost resistance from a macro perspective.

### 3.3. Comparisons with Other Methods

The expansive agent was used to improve the performance of concrete in many studies. A calcium sulfoaluminate-based expansive additive (0%, 2.5%, 5.0%, and 7.5% by the mass of the binder) was added to compensate for the shrinkage of alkali-activated material (AAM) mortar in [20]. The results show that the calcium sulfoaluminate-based expansive additive had an excellent compensation effect on the drying shrinkage of AAM mortar, but the shrinkage compensation effect was low when the modulus of elasticity was high, and thus, shrinkage stress could not be reduced from a long-term perspective. While Mohamed et al. evaluated the self-healing capability of engineered cementitious composite (ECC) produced with 5% MgO expansive agent [31]. The results show that the higher self-healing ability of ECC-MgO system was attributed to the formation of healing compounds within the crack walls because of the delayed MEA hydration, but the improvement of self-healing ability in early ages was not good. 

The results of the above studies are consistent with the research in this paper. CSA-based expansive agents can compensate for only the early shrinkage of concrete but have little effect on later-stage compensation, while MEA is characterized by delayed expansion, which has a good effect on compensating the later-stage shrinkage of mass concrete. However, the performance of concrete was improved in only a stage in their studies. A concrete with two different mix proportions of UEA and MEA was developed in our study. As a result, the performance of concrete was improved in different stages. 

It is expected that the methodologies proposed in this paper would better improve the performance of concrete. However, there is much room for further improvement. A main limiting factor is the definition of micro image and identification of products observed by SEM, because of the imperfect instrument and technology. Besides, the more mix propositions of UEA and MEA must be considered to study to determine the optimal mix proportion more accurately.

## 4. Conclusions 

In this study, testing based on rapid cycling of freezing and thawing was conducted to assess the frost resistance of concretes with double expansive source admixtures. The RDME and MLR values of various concretes were measured and SEM images of different mix designs were recorded. The conclusions are summarized as follows:(1)Based on the evolution of the RDME and MLR values, the change in concrete characteristics under cyclic freezing and thawing could be divided into three ages: a slow-damage age, a fast-damage age, and a stable age. In the slow-damage age, RDME and MLR values changed very little, with average decline and growth rates of 9.51% and 0.97%, respectively. By contrast, there were significant changes during the fast-damage age, with average decline and growth rates of 13.65% and 2.19%, respectively. As the freezing and thawing cycles continued into the stable age, the RDME and MLR values stabilized, with average decline and growth rates of 3.84% and 1.13%, respectively.(2)The concrete with UEA had a better frost resistance than other concrete during the slow-damage ages, with RDME and MLR values 17.80% higher and 74.49% lower respectively, than that of an expansive-agent-free concrete. But it rapidly lost its frost resistance during the fast-damage age, eventually losing its application properties. From a long-term perspective, the addition of double expansive agents in concrete was found to have a significant impact on the frost resistance and was associated with high RDMEs and MLRs. The frost resistance of the concrete was most improved at a UEA: MEA ratio of 2:1, with RDME and MLR values 17.35% higher and 25.1% lower respectively, than that of an expansive-agent-free concrete. The next-best results obtained at a ratio of 4:1.(3)Analysis of the concrete microstructures revealed that the internal pores of the UM_21_ were filled by large quantities of hydration products. As a result, the UM_21_ was more compact than either the UM_10_ or C and fewer microcracks appeared over the course of the freezing and thawing cycles.(4)To protect water conservancy infrastructure suffering from freezing and thawing cycles, it is suggested that a 2:1 UEA: MEA mix should be used. This mixture should effectively compensate the shrinkage and creep of the concrete and, as a result, improve the frost resistance of the hydraulic concrete and extend the service life of the infrastructure.

## Figures and Tables

**Figure 1 materials-13-01850-f001:**
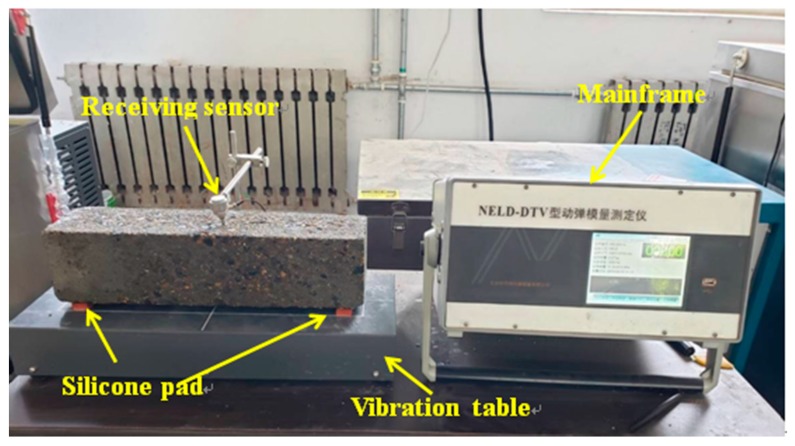
Dynamic elastic modulus tester.

**Figure 2 materials-13-01850-f002:**
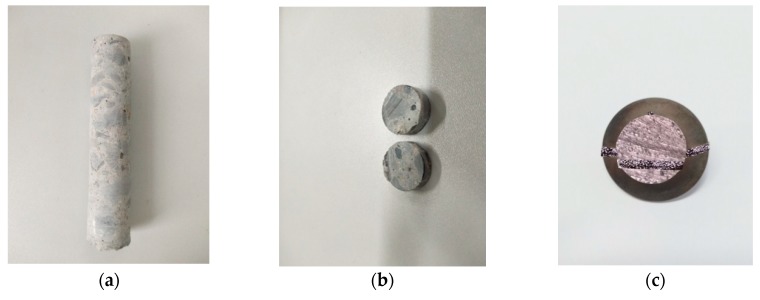
Sampling process: (**a**) drill core sample; (**b**) cut slice; (**c**) spray gold.

**Figure 3 materials-13-01850-f003:**
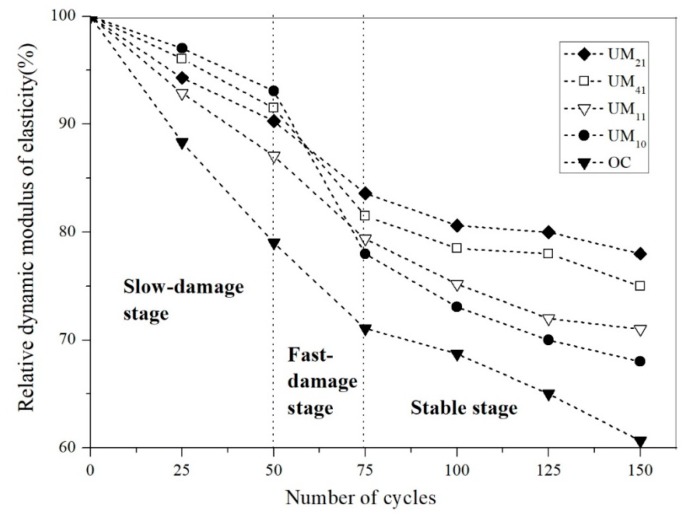
Change in relative dynamic modulus of elasticity (RDMEs) of respective concretes with increasing number of freezing and thawing cycles.

**Figure 4 materials-13-01850-f004:**
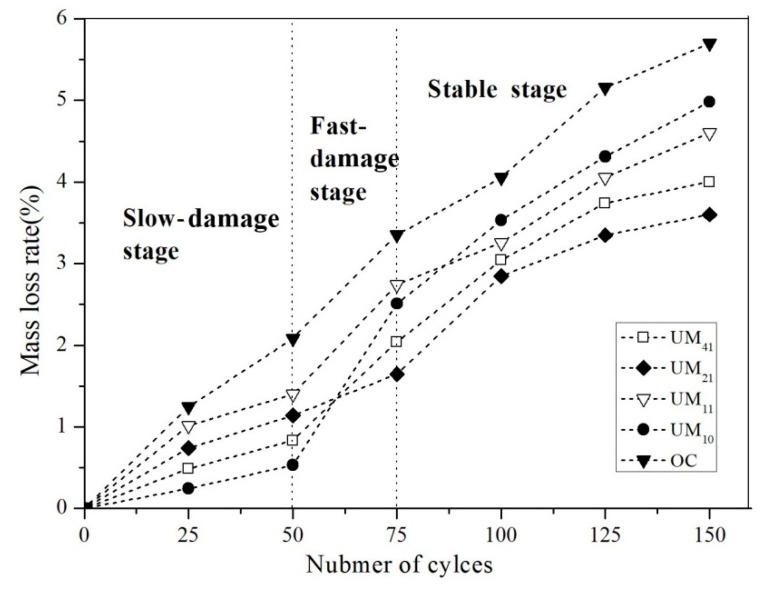
Change in mass loss rates (MLRs) of respective concretes with increasing number of freezing and thawing cycles.

**Figure 5 materials-13-01850-f005:**
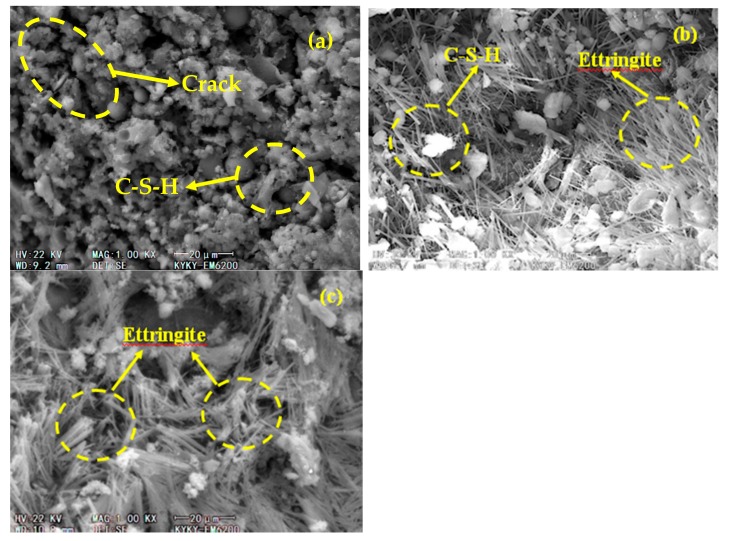
Microstructures of different concretes after 50 freezing and thawing cycles: (**a**) PC, (**b**) UM_10_, and (**c**) UM_21_.

**Figure 6 materials-13-01850-f006:**
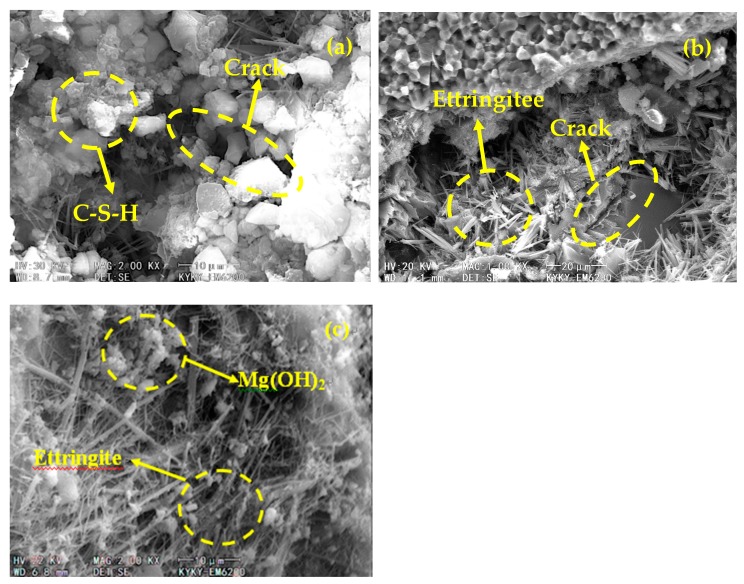
Microstructures of various concretes after 75 freezing and thawing cycles: (**a**) OC; (**b**) UM_10_; and (**c**) UM_21_.

**Figure 7 materials-13-01850-f007:**
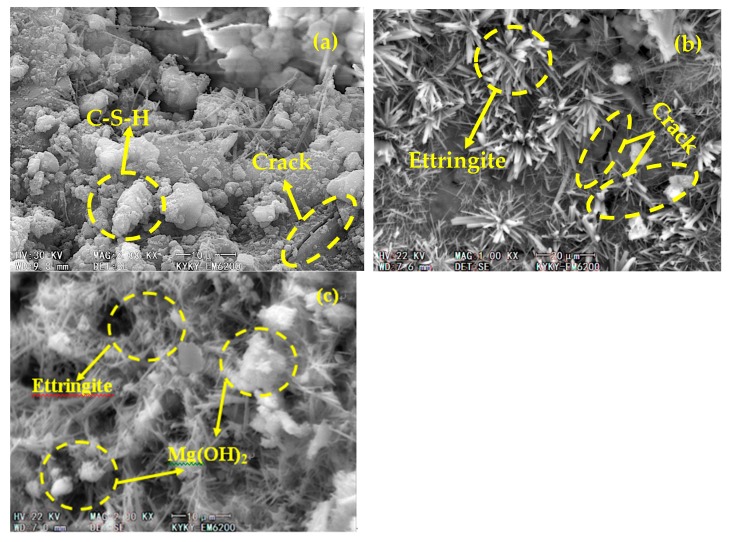
Microstructures of various concretes after 125 freezing and thawing cycles: (**a**) PC; (**b**) UM_10_; and (**c**) UM_21_.

**Table 1 materials-13-01850-t001:** Properties of the fly ashes used in this study.

Items	Fineness (%)	Water Demand Ratio (%)	Loss on Ignition (%)	SO_3_ (%)	Density (g/cm^3^)	Water Content (%)	Alkali Content (%)
Values	5.8	92	3.54	1.89	2.10	0.2	1.5

**Table 2 materials-13-01850-t002:** Properties of the superplasticizer used in this study.

Items	Water Reduction Rate (%)	Bleeding to Water (%)	Setting Time (min)		Compressive Strength		PH	Alkali Content (%)	Chloride Ion Content (%)
			Initial Setting	Final Setting	7d	28d			
Values	19	40	95	—	130	123	7.03	1.23	0.046

**Table 3 materials-13-01850-t003:** The properties of the U-type expansive agent (UEA) used in this study.

Items	Fineness		Setting Time (min)		Limited Expansion Rate	
	Specific Surface Area (m^2^/kg)	1.18-mm-square Hole Sieve Residual (%)	Initial Setting	Final Setting	7 d in Water	21 d in Air
Values	267	0	154	209	0.050	−0.010

**Table 4 materials-13-01850-t004:** Properties of MgO expansive agent (MEA) used in this study.

Items	Fineness		Setting Time (min)		Limited Expansion Rate			
	1.18-mm-square Hole Sieve Residual (%)	80-μm-square Hole Sieve Residual (%)	Initial Setting	Final Setting	7 d in 20 °C Water	Δξ in 20 °C Water	7 d in 20 °C Water	Δξ in 20 °C Water
Values	0	1.4	235	280	0.019	0.026	0.059	0.064

**Table 5 materials-13-01850-t005:** Shrinkage-compensating concrete mix designs.

Mix No.	Mass of UEA (%)	Mass of MEA (%)	Mix Proportion	Water Binder Ratio	Fly Ash (%)	Sand Ratio (%)	Water Reducing Agent (%)
UM_10_	10%	0	1:0	0.47	20%	37%	0.7%
UM_41_	8%	2%	4:1	0.47	20%	37%	0.7%
UM_21_	6.7%	3.3%	2:1	0.47	20%	37%	0.7%
UM_11_	5%	5%	1:1	0.47	20%	37%	0.7%
PC	0	0	--	0.47	20%	37%	0.7%

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
