# Peer review of "Experimental Study on Freezing and Thawing Cycles of Shrinkage-Compensating Concrete with Double Expansive Agents"

_materials, 2020, doi:10.3390/ma13081850_

Round 1

Reviewer 1 Report

Research is interesting, however, in my opinion, it requires some explanations and corrections.

My main concern is the usage of expansive agents that forms Candlot salt which is known from its corrosive acting in concrete. It fills the pores and is strengthening the concrete in the first stage and deteriorates concrete after the tension pressure is exceeded. It should be described and explained.  

L110: GB175-2007 is not referenced

Table 2, 3, 4 - what are the units of setting time?

L146-163: what is the reference document. Is it [18]? It is not clear enough.

Figure 2. It would be good to describe it in a similar form as fig. 1

L167: ...with "an SEM" - shortcut should be explained, "an" is not the proper world.

Literature review could be more detailed.

Yours Sincerelly,

Reviewer.

Reviewer 2 Report

The manuscript entitled “Experimental study on freezing and thawing cycles of shrinkage-compensating concrete with double expansive agents” investigates the effect of double expansive admixture on frost resistance of concrete. However, the paper needs to be strengthened to be published in Materials.

  • The manuscript needs to be proofread.
  • Line 20-23 needs revision
  • Lin 42: “In the United States, the annual economic loss caused by frost damage is more than 6 billion dollars, while in northeast China nearly 100% of concrete buildings have been damaged by freezing and thawing.” Needs a citation
  • Line 66: please mere the citations (11 and 12)
  • Line 75, 77: Please check the referencing style to be matched with Materials format.
  • Table 1: please use the units in parenthesis; same comments for the other tables
  • Equations 1 and 2 needs citation
  • Figure 2 must be removed as it doesn’t provide any information/value
  • Since SEM was used to observe CHS products, the samples must be prepared in a special procedure. What procedure the authors used to prepare a flat sample?
  • What was the working condition of SEM imaging?
  • Figures 3 and 4 must be updated (there are few blue lines wound the figures)
  • Line 185: how the authors distinguish three phases?
  • Identification of three phases is not the outcome from the frost resistance of concrete (section 3.1). The authors must interpret the results obtained in the graphs.
  • Fig 5a-7b: to me the red dotted line is not a crack
  • Fig 5a: how the author identified CSH products? Have you performed EDS?
  • The paper lacks a good discussion on why such results have been obtained and compare those results with literature

Reviewer 3 Report

This work experimentally investigated the efficiency of using double expansive agents (i.e. UEA and MEA) to increase the frost resistance of concrete. Though few interesting results are presented, this work is not mutual enough at this state to be published. Major issues are listed as followed:

  1. The experimental program is not properly proposed, which does not allow to conclude on the coupled effects of UEA and MEA.
  2. Lines 42-49: Why did you only discuss for the case of China and USA?  Citations are also needed.
  3. Line 65: cementitious materials systems ---> supplementary cementitious materials
  4. Table 2: How do you determine setting times of SP?
  5. Tables 3 and 4: again, what does it mean by setting times of UEA and MEA?
  6. Concrete mix design: The authors should justify why such compositions, UEA:MEA ratios are used?
  7. Line 168: C-H-S --> C-S-H. How do you distinguish Mg(OH)2 and AFt?    
  8. Fig. 2: no sense to show the SEM microscope
  9. 3.1 Analysis of frost resistance of concrete: It’s not justified and dangerous when the authors try to align the experimental data to the common 3 states of damaging due to free-thaw cycling. Fig. 3 and especially Fig. 4 do not prove this at all. Therefore the following interpretation (sections 3.1.1 -> 3.1.3) does not make sense.  
  10. Lines 200-201: based on what you can conclude that 0.97% mass loss indicate a good frost resistance?   
  11. Lines 220-221: you did not know if AFt is formed or not because you have not shown SEM image, so you cannot make such statement. Better to  analyze SEM images first.
  12. Lines 232-233: how do you say this?
  13. All SEM: except for AFt, how do you indicate different phases without EDX?
  14. The delayed formation of Mg(OH)2 or AFt could cause cracking of concrete, did you consider this in your study?
  15. Conclusion 4: it’s too little dataset to conclude that.

Reviewer 4 Report

This paper describes experimental considerations on improving the frost resistance of expanded concrete using various expanding agents.

It is considered that there is no clear correlation between the shrinkage characteristics of concrete and the frost damage resistance.

On the other hand, concrete solidity (water tightness) should have a strong relationship with frost damage resistance.

On the other hand, there must be a strong relationship between the concreteness (water tightness) of concrete and the frost resistance. If the action of the expansive agent increases the solidity, the concrete must be restrained from the surroundings while it hardens.

Why did the relative dynamic elastic modulus of the concrete specimen be derived from the resonance frequency of the flexural vibration (guessed from Fig. 1)?

Round 2

Reviewer 1 Report

Dear Authors

My previous concerns have been revised, thank you.

L189-195: Please, try to describe the procedure in a clearer form.

Figure 2 (c): what is shown on the figure? "attach conductive film" seems not a proper caption.

Figure 4: Should be "Fast-damage stage" not "First-damage stage"

L200+: On the figures nomenclature, "fast, slow, stable damage stage" is being used. In the text, it is "fast, slow, stable damage age". It would be good to use consistent nomenclature or to clarify this.

L147: OC shortcut is described as "plain concrete". In my understanding "PC is concrete without reinforcement. OC should be described as "ordinary concrete".

Congratulations on your work!

Reviewer 2 Report

The authors carefully addressed all comments and therefore I support for publication.

Author Response

Sincerely thank you very much for your valuable comments and your approval.

Reviewer 3 Report

The authors have addressed most of the reviewers comments, quality of manuscript is relatively improved. My comments on 3 stages damage is still remained, the authors should think over this issue. Furthermore, language should be improved. Figures should be reorganized/combined to save space and provide concise/relevant information.        

Reviewer 4 Report

In abstract, don't use abbreviated phrases; UEA and MEA. What are the UEA and MEA?

Correct Eq. (2) as MEA: MgO + H2O→Mg(OH)2

Complete Table 5. There is no aggregate.

The author mentioned, "The freezing and thawing cycles test was generally used to simulate the damage to buildings...". Also, the author indicated the relation between the cracking by shrinkage and the durability for freezing and thawing affection. However, the experiment was performed in accordance with a method based on a standard freeze-thaw test, and did not cause cracks during shrinkage, nor was the specimen having a reinforcing bar as an external constraint required for the shrinkage. Even, mention nothing about the effects of shrinkage, but just mention only that using a couple of expanding agents to improve the solidity would increase resistance to freeze-thaw action.

Round 3

Reviewer 1 Report

Dear Authors

L189-195 in the revised manuscript version 2 does not describe the core drilling process as it has been mentioned in the response (it is described in lines 185 - 189 there).

L189-200 is still not clearly described. Very poor English is used in the corrected fragments of text, f.e. it is written:

  • "Firstly, the thin films were put into a dry box for drying." what kind of thin films were dried, should it be the cut slices of concrete? It is obvious that samples are put in the drying machine for drying. What moisture condition of samples has been achieved in this process?
  • "a platinum spray was used to apply a coat... which was then attached to a copper production with conductive adhesive..."
  • "The processed samples were used to observed. And the observation must be carried out in a closed space without air."

Figure 2c is not showing the process, it is showing the machine. It is no sense to show the small figure of machine used for the coating process.

Figure 3 has been corrected from "first damage stage" to "fast damge stage" What does it mean "damge"?

The new fragments of text contain a lot of language errors and ambiguities.

English language of corrected fragments of text and figures needs proofreading.

Yours Sincerely,

Reviewer.

Reviewer 3 Report

The authors have addressed all comments of the reviewers. I recommend for publication at this state.

Author Response

(The authors gave the same response as above.)
